# Gamma activity accelerates during prefrontal development

**Sebastian H Bitzenhofer[†]\*, Jastyn A Pöpplau, Ileana Hanganu-Opatz\***

Institute of Developmental Neurophysiology, Center for Molecular Neurobiology, University Medical Center Hamburg-Eppendorf, Hamburg, Germany

**Abstract** Gamma oscillations are a prominent activity pattern in the cerebral cortex. While gamma rhythms have been extensively studied in the adult prefrontal cortex in the context of cognitive (dys)functions, little is known about their development. We addressed this issue by using extracellular recordings and optogenetic stimulations in mice across postnatal development. We show that fast rhythmic activity in the prefrontal cortex becomes prominent during the second postnatal week. While initially at about 15 Hz, fast oscillatory activity progressively accelerates with age and stabilizes within gamma frequency range (30–80 Hz) during the fourth postnatal week. Activation of layer 2/3 pyramidal neurons drives fast oscillations throughout development, yet the acceleration of their frequency follows similar temporal dynamics as the maturation of fast-spiking interneurons. These findings uncover the development of prefrontal gamma activity and provide a framework to examine the origin of abnormal gamma activity in neurodevelopmental disorders.

**\*For correspondence:**
sbitzenhofer@ucsd.edu (SHB);
ileana.hanganu-opatz@zmnh.uni-hamburg.de (IH-O)

**Present address:** [†]Center for Neural Circuits and Behavior, Department of Neurosciences, University of California, San Diego, United States

**Competing interests:** The authors declare that no competing interests exist.

## Introduction

Synchronization of neuronal activity in fast oscillatory rhythms is a commonly observed feature in the adult cerebral cortex. While its exact functions are still a matter of debate, oscillatory activity in gamma frequency range has been proposed to organize neuronal ensembles and to shape information processing in cortical networks (*Singer, 2018*; *Cardin, 2016*; *Sohal, 2016*). Gamma activity emerges from reciprocal interactions between excitatory and inhibitory neurons. In the visual cortex, fast inhibitory feedback via soma-targeting parvalbumin (PV)-expressing inhibitory interneurons leads to fast gamma activity (30–80 Hz) (*Cardin et al., 2009*; *Chen et al., 2017*), whereas dendrite-targeting somatostatin (SOM)-expressing inhibitory interneurons contribute to beta/low gamma activity (20–40 Hz) (*Chen et al., 2017*; *Veit et al., 2017*). A fine-tuned balance between excitatory drive and inhibitory feedback is mandatory for circuit function underlying cognitive performance. Interneuronal dysfunction and ensuing abnormal gamma activity in the medial prefrontal cortex (mPFC) have been linked to impaired cognitive flexibility (*Cho et al., 2015*). Moreover, imbalance between excitation and inhibition in cortical networks and resulting gamma disruption have been proposed to cause cognitive disabilities in autism and schizophrenia (*Cho et al., 2015*; *Cao et al., 2018*; *Rojas and Wilson, 2014*).

Despite substantial literature linking cognitive abilities and disabilities to gamma oscillations in the adult mPFC, the ontogeny of prefrontal gamma activity is poorly understood. This knowledge gap is even more striking considering that abnormal patterns of fast oscillatory activity have been described at early postnatal age in autism and schizophrenia mouse models (*Chini et al., 2020*; *Richter et al., 2019*; *Hartung et al., 2016*). Knowing the time course of prefrontal gamma maturation is essential for understanding the developmental aspects of mental disorders.

To this end, we performed an in-depth investigation of the developmental profile of gamma activity in the mouse mPFC from postnatal day (P) five until P40. We show that pronounced fast oscillatory activity emerges toward the end of the second postnatal week and increases in frequency and amplitude with age. While activation of layer 2/3 pyramidal neurons (L2/3 PYRs) drives fast

oscillatory activity throughout development, the acceleration of its frequency follows the same dynamics as the maturation of inhibitory feedback and fast-spiking (FS) interneurons.

## Results

### Fast oscillatory activity in the prefrontal cortex accelerates during development

Extracellular recordings in the mPFC of anesthetized and non-anesthetized P5-40 mice revealed that oscillatory activity at fast frequencies (>12 Hz) can be detected at the beginning of the second postnatal week. The temporal dynamics of these fast oscillations are similar in the two states, yet their magnitude is higher in non-anesthetized mice, as previously described (*Chini et al., 2019*). The magnitude of fast oscillations increases with age (Mann-Kendall trend test, $p=3.93\times10^{-22}$, n = 114 recordings, tau-b 0.625) and can be detected as distinct peaks in power spectra at the end of the second postnatal week (*Figure 1a,b*). The peak frequency of these oscillations gradually increases with age (Mann-Kendall trend test, $p=2.73\times10^{-8}$, n = 114 recordings, tau-b 0.361), starting at ~ 20 Hz at P12 and reaching the characteristic gamma frequency of 50–60 Hz at P25 (*Figure 1b–d*). Both, peak strength and peak frequency, do not change after P25. A linear regression model of peak frequency and peak amplitude shows significant correlation with age (n = 114, df = 111, $R^2$ = 0.542, $p=5.48\times10^{-20}$; ANOVA: peak frequency $F(1,111)=17.8$, $p=4.86\times10^{-5}$, peak amplitude $F(1,111)=74.4$, $p=5.05\times10^{-14}$).

### FS interneuron maturation resembles the time course of gamma development

FS PV-expressing interneurons have been identified as key elements for the generation of oscillatory activity in gamma frequency range in the adult cortex (*Cardin et al., 2009*). To assess whether the developmental gamma dynamics relate to FS PV-expressing interneuron maturation, we performed immunohistochemistry and single unit analysis in P5-40 mice.

First, immunostainings showed that PV expression in the mPFC starts at the end of the second postnatal week, increases until P25 and stabilizes afterwards (Mann-Kendall trend test, $p=1.29\times10^{-7}$, n = 38 mice, tau-b 0.623) (*Figure 2a*). These dynamics over age follow a similar trend as the changes in peak power and peak frequency of the fast oscillations described above. In

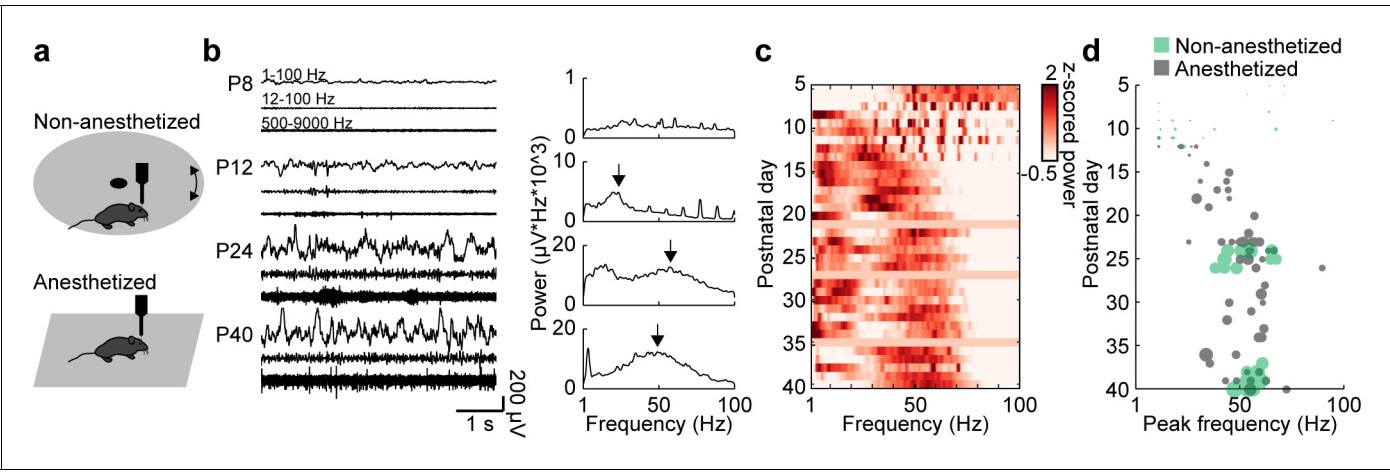

**Figure 1.** Development of gamma activity in the mouse mPFC. (a) Schematic of extracellular recordings in the mPFC of anesthetized and non-anesthetized P5-40 mice. (b) Characteristic examples of extracellular recordings of local field potentials (LFP) and multi-unit activity (MUA) at different ages after band-pass filtering (left) and the corresponding power spectra (right). (c) Z-scored average power spectra of spontaneous oscillatory activity for P5-40 mice (n = 114 recordings from 100 mice). (d) Scatter plot displaying peak frequencies of fast oscillations (12–100 Hz) during postnatal development of anesthetized (gray, n = 80 recordings/mice) and non-anesthetized mice (green, n = 34 recordings from 20 mice). Marker size displays peak strength. (See *Supplementary file 1* for a summary of experimental conditions. See *Supplementary file 2* for statistics.).

The online version of this article includes the following source data for figure 1:

**Source data 1.** Source data for *Figure 1b*.

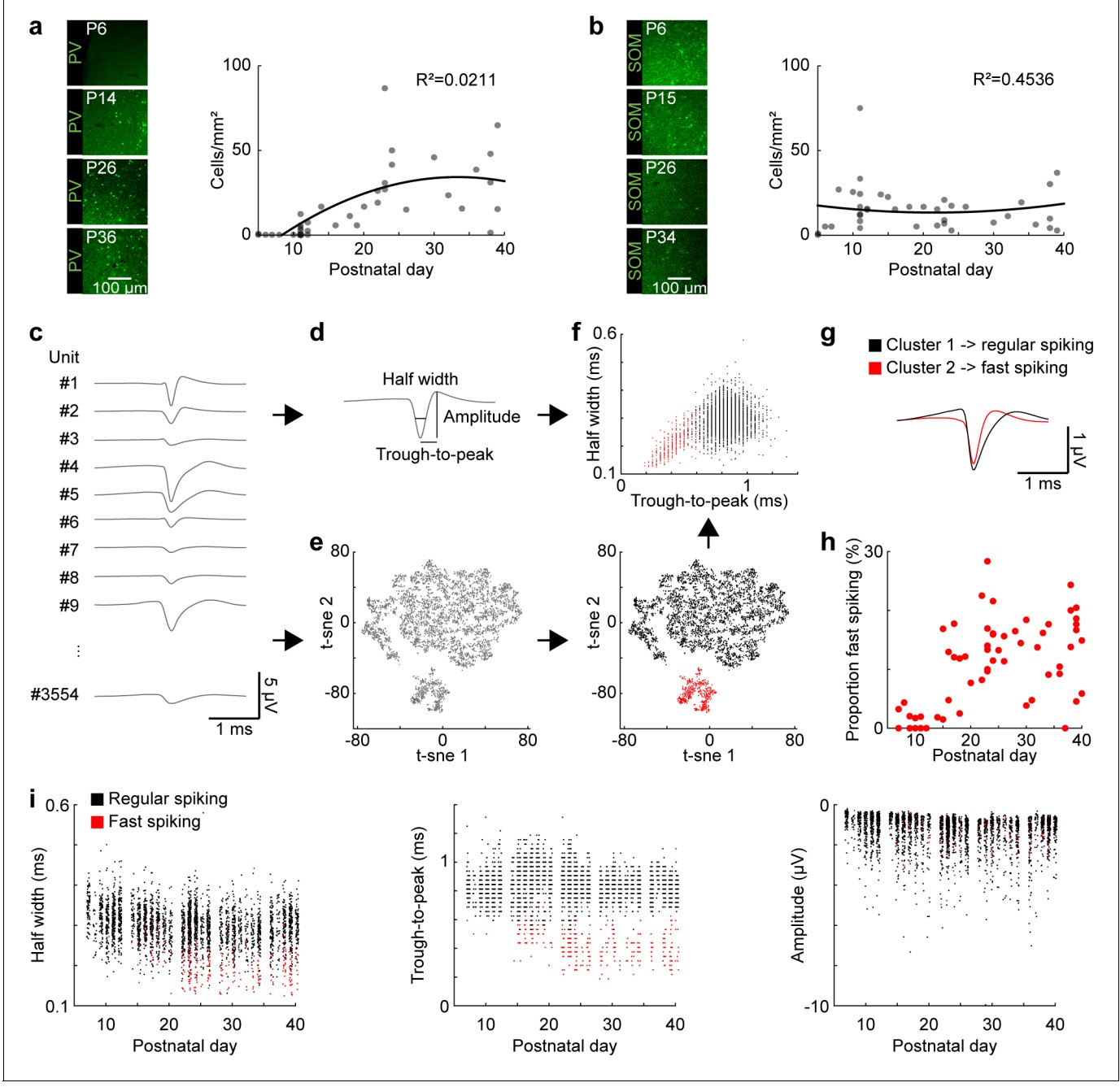

**Figure 2.** Development of FS interneurons in the mouse mPFC. (a) Left, examples of PV immunostaining in the mPFC at different ages. Right, scatter plot displaying the density of PV-immunopositive neurons in the mPFC of P5-40 mice (n = 38 mice). (b) Same as (a) for SOM-immunopositive neurons (n = 39 mice). (c) Example mean waveforms of extracellular recorded single units from P5-40 mice. (d) Schematic showing features classically used to distinguish RS and FS units in adult mice. (e) Left, scatter plot showing the first two components of a t-sne dimensionality reduction on the mean waveforms for all units recorded from P5-40 mice (n = 3554 units from 66 recordings/mice). Right, same as left with the first two clusters obtained by hierarchical clustering labeled in black and red. (f) Scatter plot of half width and trough-to-peak time for cluster 1 (black) and 2 (red). (g) Mean waveform for cluster 1 (black) and 2 (red). (h) Scatter plot showing the proportion of FS units for P5-40 mice. (i) Scatter plots showing classic spike shape features for P5-40 for cluster 1 (RS, black) and 2 (FS, red). (See *Supplementary file 1* for a summary of experimental conditions. See *Supplementary file 2* for statistics).

The online version of this article includes the following source data and figure supplement(s) for figure 2:

**Source data 1.** Source data for *Figure 2a,b*.
**Source data 2.** Source data for *Figure 2f*.
**Source data 3.** Source data for *Figure 2g*.

*Figure 2 continued on next page*

*Figure 2 continued*

**Source data 4.** Source data for *Figure 2h*.
**Source data 5.** Source data for *Figure 2i*.
**Figure supplement 1.** t-sne dimensionality reduction.
**Figure supplement 1—source data 1.** Source data for *Figure 2—figure supplement 1b–d*.

contrast, the number of SOM positive neurons does not significantly vary along postnatal development (Mann-Kendall trend test, p=0.99, n = 39 mice, tau-b −0.003) (*Figure 2b*).

Second, to directly assess the functional maturation of FS putatively PV-expressing neurons, we used bilateral extracellular recordings and identified single unit activity (SUA). The classically used action potential features to distinguish adult FS and regular-spiking (RS) neurons (i.e. trough to peak time and half width) cannot be applied during early development, because of a strong overlap of these features. Therefore, we developed an algorithm to classify RS and FS units using dimensionality reduction with t-Distributed Stochastic Neighbor Embedding (t-sne) on the mean waveforms of all units recorded across development, followed by hierarchical clustering based on pairwise Euclidean distance (*Figure 2c–e*). This approach resulted in an unbiased detection of FS units across age (*Figure 2f,g*). Of note, a more detailed analysis revealed that the dimensionality reduction based on mean waveforms correlates with features like trough to peak and half width, but is less affected by age, cortical layer, and amplitude (*Figure 2—figure supplement 1*). The classification of units with this method revealed that FS units start to be detected at the end of the second postnatal week. Their number gradually increased until P25 (Mann-Kendall trend test, p=1.44×10⁻⁷, n = 66 recordings, tau-b 0.458) (*Figure 2h*). A comparison of the classical features across age showed that trough-to-peak duration (Mann-Kendall trend test, RS, p=4.57×10⁻⁵, n = 3172 units, tau-b −0.051; FS, p=9.23×10⁻¹¹, n = 382 units, tau-b −0.236), half width (Mann-Kendall trend test, RS, p=1.61×10⁻²⁸, n = 3172 units, tau-b −0.134; FS, p=5.17×10⁻¹⁷, n = 382 units, tau-b −0.295), and negative amplitude (Mann-Kendall trend test, RS, p=4.45×10⁻²², n = 3172 units, tau-b −0.117; FS, p=3.82×10⁻⁶, n = 382 units, tau-b −0.163) of RS and FS units gradually decreased from the end of the second

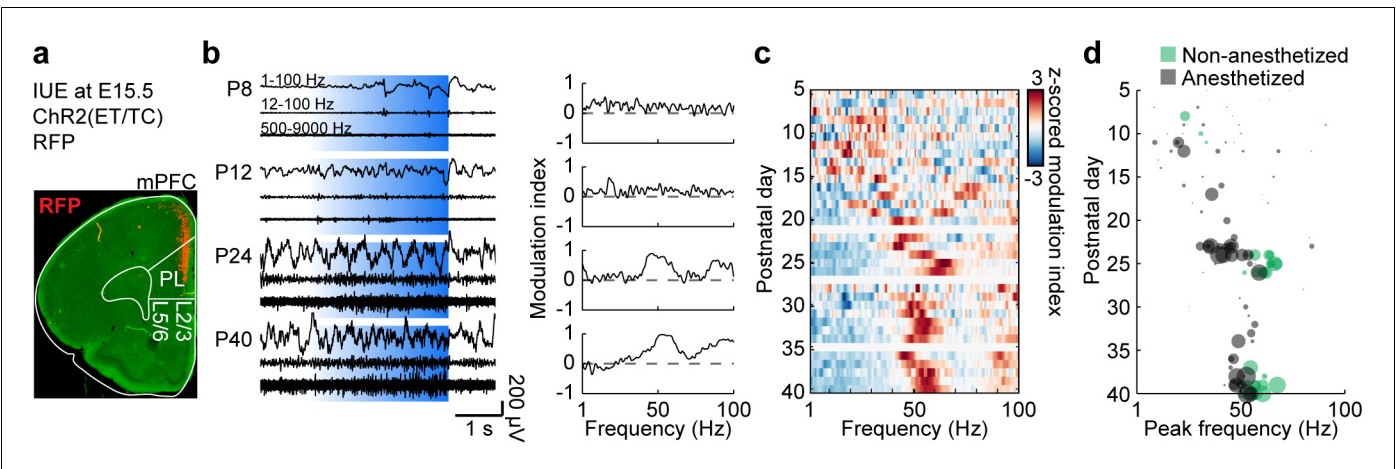

**Figure 3.** Development of L2/3 PYR-driven gamma in the mPFC. (**a**) ChR2(ET/TC)−2A-RFP-expression in L2/3 PYRs in mPFC after IUE at E15.5 in a coronal slice of a P10 mouse. (**b**) Characteristic examples of extracellular recordings of LFP and MUA during ramp light stimulations (473 nm, 3 s) of prefrontal L2/3 PYRs at different ages (left) and the corresponding MI of power spectra (right). (**c**) Z-scored average MI of power spectra for P5-40 mice (n = 115 recordings from 101 mice). (**d**) Scatter plot displaying stimulus induced peak frequencies during postnatal development for anesthetized (gray, n = 80 recordings/mice) and non-anesthetized mice (green, n = 35 recordings from 21 mice). Marker size displays peak strength. (See *Supplementary file 1* for a summary of experimental conditions. See *Supplementary file 2* for statistics).
The online version of this article includes the following source data and figure supplement(s) for figure 3:

**Source data 1.** Source data for *Figure 3d*.
**Figure supplement 1.** Control stimulations of L2/3 PYRs in the mPFC.
**Figure supplement 1—source data 1.** Source data for *Figure 3—figure supplement 1c*.

postnatal week until P25. However, the most prominent changes were detected for through-to-peak duration and half width of FS units (*Figure 2i*). A linear regression model of single unit features shows significant correlation with age (n = 52, df = 44, $R^2$ = 0.550, p=$2.31\times10^{-7}$; see *Supplementary file 2* for details). These results are consistent with a detailed description of the physiological development of prefrontal PV-expressing interneurons performed in brain slices (*Miyamae et al., 2017*).

Thus, in line with the immunohistochemical examination, the analysis of single units showed that FS putatively PV-positive interneurons show similar dynamics of maturation as fast oscillations recorded in the mPFC of P5-40 mice.

## Activation of L2/3 pyramidal neurons drives fast oscillations with similar acceleration across development as spontaneous activity

Besides FS interneurons, L2/3 PYRs in mPFC have been found to induce fast oscillations in the mPFC of P8-10 mice. Their non-rhythmic activation (but not activation of L5/6PYRs) drives oscillatory activity peaking within 15–20 Hz range (*Bitzenhofer et al., 2017a*; *Bitzenhofer et al., 2017b*), similar to the peak frequency of spontaneous network activity at this age. To test if L2/3 PYRs-driven activity also accelerates with age, we optogenetically manipulated these neurons in P5-40 mice. Stable expression of the light-sensitive channelrhodopsin two derivate E123T T159C (ChR2(ET/TC)) restricted to about 25% of PYR in L2/3 of the mPFC was achieved by in utero electroporation (IUE) at embryonic day (E) 15.5 (*Figure 3a*). Optogenetic stimulation with ramps of steadily increasing light power (473 nm, 3 s, 30 repetitions) were performed during extracellular recordings in the mPFC. As previously shown, this type of stimulation activates the network without forcing a specific rhythm (*Bitzenhofer et al., 2017a*).

Similar to spontaneous activity, activating L2/3 PYRs induced oscillatory activity with a gradually increasing frequency during development (*Figure 3b–d*). Consistent peaks in the modulation index (MI) of power spectra were detected at 15–20 Hz at the beginning of the second postnatal week and increased in frequency (Mann-Kendall trend test, p=$7.69\times10^{-6}$, n = 115 recordings, tau-b 0.288) and amplitude (Mann-Kendall trend test, p=$1.04\times10^{-9}$, n = 115 recordings, tau-b 0.392) until reaching stable values within 50–60 Hz at P25. A linear regression model of peak frequency and peak amplitude shows significant correlation with age (n = 115, df = 112, $R^2$ = 0.364, p=$3.72\times10^{-12}$; ANOVA: peak frequency $F_{(1,112)}$=13.9, p=$2.95\times10^{-4}$, peak amplitude $F_{(1,112)}$=43.8, p=$1.26\times10^{-9}$). Control stimulations with light that does not activate ChR2(ET/TC) (594 nm, 3 s, 30 repetitions) did not induce activity and led to the detection of unspecific peak frequencies (Mann-Kendall trend test, p=0.09, n = 111 recordings, tau-b 0.111) and low amplitudes (Mann-Kendall trend test, p=0.74, n = 111 recordings, tau-b 0.022) (*Figure 3—figure supplement 1*).

Peak frequency and amplitude of activity induced by ramp light stimulation are significantly correlated with peak frequency and amplitude of spontaneous activity at the level of individual recordings (peak frequency n = 114, df = 112, $R^2$ = 0.16, p=$6.00\times10^{-6}$; peak amplitude n = 114, df = 112, $R^2$ = 0.206, p=$2.30\times10^{-7}$). Thus, L2/3 PYR-driven activity in the mPFC follows the same developmental dynamics as spontaneous activity indicating the importance of L2/3 PYRs for gamma maturation.

## The rhythmicity of pyramidal cell and interneuron firing follows similar development as accelerating gamma activity in mPFC

To assess the contribution of distinct cell types to the emergence of gamma during postnatal development, we compared the firing of RS units, mainly corresponding to PYRs, and FS units, mainly corresponding to PV-expressing interneurons, during ramp light stimulations of L2/3 PYRs in P5-40 mice (*Figure 4a,b*, *Figure 4—figure supplement 1*).

The average firing rate of RS and FS units in the stimulated hemisphere in the mPFC increased in response to ramp stimulation (*Figure 4c*). While ramp-induced firing rate changes averaged for RS units (Mann-Kendall trend test, p=0.07, n = 7 age groups, tau-b 0.619) became more prominent at older age, the average firing rate changes were stable for FS units (Mann-Kendall trend test, p=0.88, n = 7 age groups, tau-b 0.047) (*Figure 4d*). At the level of individual units, most RS units showed positive modulation of their firing rates in response to stimulation at P5-10, whereas at older age the proportion of positively modulated units decreased (Mann-Kendall trend test, p=$1.52\times10^{-14}$,

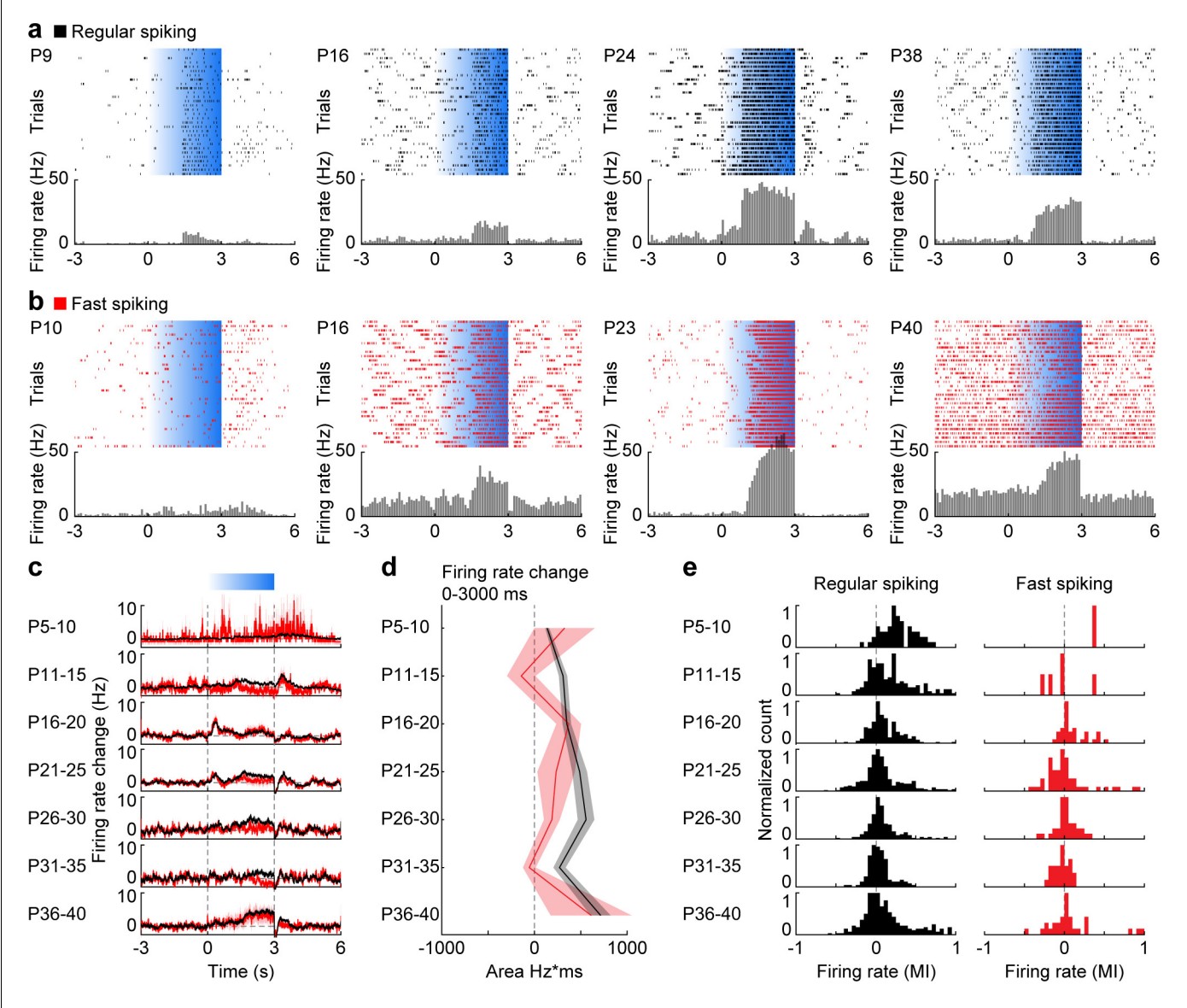

**Figure 4.** Development of RS and FS unit activity during L2/3 PYR-driven gamma in the mPFC. (a) Raster plots and peri-stimulus time histograms for activated RS example units in response to ramp light stimulation (3 s, 473 nm, 30 repetitions) of prefrontal PYRs at different ages. (b) Same as (a) for FS units. (c) Average firing rate change of RS (black, n = 1824 units from 66 recordings/mice) and FS (red, n = 226 units from 66 recordings/mice) units in response to ramp light stimulation of prefrontal L2/3 PYRs for different age groups. (d) Line plot displaying the average firing rate changes of RS and FS units during ramp light stimulation for different age groups. (e) Histograms of the MI of firing rates in response to ramp light stimulation for RS and FS units. (Average data is displayed as mean ± sem. See *Supplementary file 1* for a summary of experimental conditions. See *Supplementary file 2* for statistics.).

The online version of this article includes the following source data and figure supplement(s) for figure 4:

**Source data 1.** Source data for *Figure 4d*.

**Figure supplement 1.** RS and FS unit activity during L2/3 PYR-driven gamma in the mPFC.

n = 1821 units, tau-b −0.123) (*Figure 4e*). In contrast, individual FS units showed both positive and negative modulation of activity throughout development (Mann-Kendall trend test, p=0.91, n = 225 units, tau-b −0.005), yet the low number of FS units at young age precluded clear conclusions. Thus, during early postnatal development most RS units are activated by ramp light stimulations but only moderately increase their firing rate. During late postnatal development some RS units strongly increase their firing rate, whereas others reduce their firing rate.

Next, we tested whether RS and FS units engage in rhythmic activity and calculated autocorrelations and spike-triggered LFP power of individual units. While no clear peaks of rhythmicity were found during spontaneous activity before stimulations (*Figure 5—figure supplement 1*), autocorrelations showed that a subset of RS and FS units fire rhythmically in response to ramp light stimulation of prefrontal L2/3 PYRs (*Figure 5a*). The power of autocorrelations revealed that prominent rhythmic firing starts at about P15 and increases in frequency before it stabilizes at about P25 (*Figure 5b*). A multifactorial ANOVA shows significant effects for age group ($F_{(6,153093)}$ = 25.4, p=$2.21 \times 10^{-30}$) and frequency ($F_{(99,153093)}$ = 492.7, p=0.000), but not between RS and FS units ($F_{(1,153093)}$ = 1.56, p=0.211). Next, we calculated the power of averaged spike-triggered LFPs to

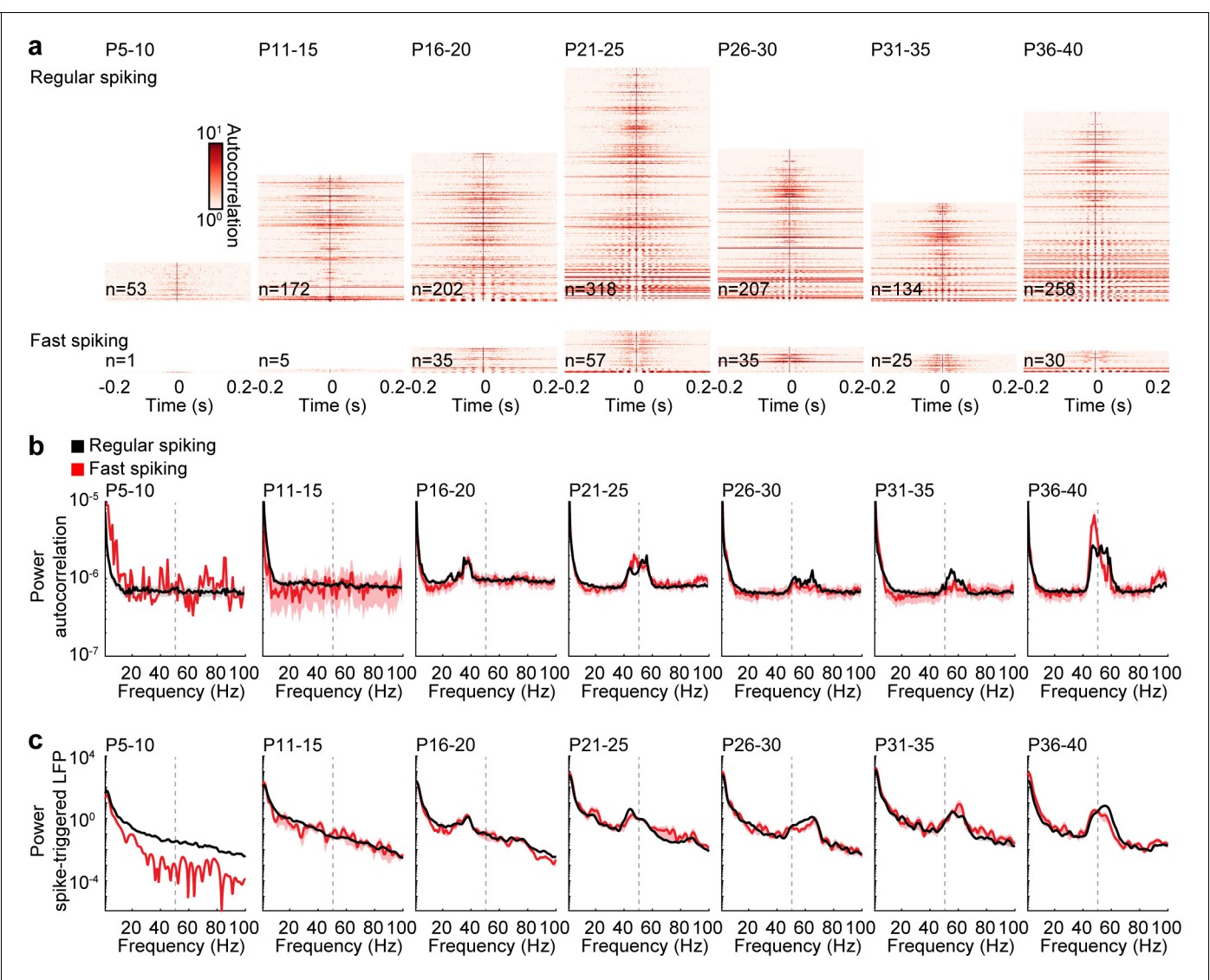

**Figure 5.** Rhythmicity of RS and FS units across age. (**a**) Color-coded autocorrelations of prefrontal RS (top) and FS (bottom) units during ramp light stimulation (3 s, 473 nm) for different age groups. Each row represents one unit (only units firing > 1 Hz are included). (**b**) Average autocorrelation power of RS (black) and FS (red) units during ramp light stimulation for different age groups. (**c**) Average power of mean spike-triggered LFP of RS (black) and FS (red) units during ramp light stimulation for different age groups. (Average data is displayed as mean ± sem. See *Supplementary file 1* for a summary of experimental conditions. See *Supplementary file 2* for statistics.).

The online version of this article includes the following figure supplement(s) for figure 5:

**Figure supplement 1.** Rhythmicity of RS and FS units during spontaneous activity.

**Figure supplement 2.** Additional measures for rhythmicity of RS and FS units.

**Figure supplement 3.** Crosscorrelations of RS and FS units during spontaneous activity.

examine the interaction of single unit rhythmicity with oscillatory LFP activity. The development of spike-triggered LFP power is consistent with the development of single unit autocorrelation power (multifactorial ANOVA: unit type $F_{(1,814825)}$ = 277.2, p=3.12×10$^{-62}$, age group $F_{(6,814825)}$ = 847.4, p=0.000, frequency $F_{(400, 814825)}$=614.5, p=0.000), indicating that single unit rhythmicity of local neurons is reflected in the prefrontal LFP. Consistent results were obtained for inter-spike intervals and pairwise phase consistency (*Figure 5—figure supplement 2*), as well as crosscorrelations between simultaneously recorded unit pairs (*Figure 5—figure supplement 3*). RS units show higher values at lower instantaneous frequencies for inter-spike intervals than FS units, suggesting that they tend to skip cycles of induced gamma activity (*Figure 5—figure supplement 2d*). Overall, the dynamics of RS and FS rhythmicity are similar to the development of spontaneous and stimulated gamma activity, indicating close interactions between RS and FS units during fast oscillations.

## Inhibitory feedback maturation resembles the dynamics of gamma development

Stimulation of prefrontal L2/3 PYRs with short light pulses (3 ms, 473 nm) at different frequencies was used to test the maximal firing frequencies of RS and FS units in P5-40 mice. Pulse stimulations induced a short increase of firing for both RS and FS units (*Figure 6a*). Confirming previous results (*Bitzenhofer et al., 2017a*), RS units did not follow high stimulation frequencies in mice younger than P11 and showed strong attenuation in the response to repetitive pulses. With ongoing development, this attenuation at high stimulation frequencies became less prominent for RS and FS units (*Figure 6b*), yet the low number of FS units at young age precluded clear conclusions. Inter-spike intervals of individual units revealed several peaks at fractions of the stimulation frequency for RS and FS, especially at higher stimulation frequencies (*Figure 6—figure supplement 1*). These data suggest that individual units do not fire in response to every light pulse in the pulse train but skip some pulses.

To assess the development of inhibitory feedback, we examined the firing rate changes of RS and FS units from P5-40 mice in response to individual 3 ms-long light pulses. The firing rate of RS and FS units transiently increased after pulse stimulation (*Figure 6c*). This effect significantly increased with age for RS units (Mann-Kendall trend test, p=0.04, n = 7 age groups, tau-b 0.714), but not for FS units (Mann-Kendall trend test, p=0.07, n = 7 age groups, tau-b 0.619) (*Figure 6d*). Next, we analyzed the delays of light-induced firing peaks for the two populations of units. The similar delays observed for RS and FS units suggest that the majority of RS units are non-transfected neurons, that are indirectly activated. The initial peak of increased firing was followed by reduced firing rates for RS and FS units only during late postnatal development (*Figure 6e,f*). The magnitude and duration of this firing depression gradually augmented with age and reached significance for RS units (Mann-Kendall trend test, RS, p=6.9×10$^{-3}$, n = 7 age groups, tau-b −0.905; FS, p=0.07, n = 7 age groups, tau-b −0.619). Thus, the inhibitory feedback in response to L2/3 PYR firing in the mPFC increases with age.

## FS characteristics and rhythmicity of single units relate to the frequency of fast oscillatory activity

We used stepwise linear regression models to identify the most important correlations of single unit measures with gamma activity across development at the level of individual mice. The models included the following predictor variables (average values for each mouse): (i) proportion of FS units, action potential half width, trough-to-peak, and amplitude for RS and FS units (see *Figure 2*), (ii) firing rate change, peak power of autocorrelation, and peak power of spike-triggered LFP for RS and FS units during ramp light stimulation (see *Figure 5*), and (iii) firing rate change for RS and FS units 0–10 ms and 10–50 ms after pulse stimulation (see *Figure 6*). Stepwise linear regression to predict the peak frequency during ramp light stimulation identified action potential half width of FS units, as well as peak power of autocorrelation of RS and FS units as best predictor variables (n = 42, df = 38, $R^2$ = 0.136, p=0.036; see *Supplementary file 2* for detailed statistics). Stepwise linear regression to predict the peak amplitude during ramp light stimulation identified firing rate change of RS units during ramp light stimulation, peak power of spike-triggered LFP for RS units, as well as RS firing rate change 0–10 ms after pulse stimulation and FS firing rate change 10–50 ms after pulse

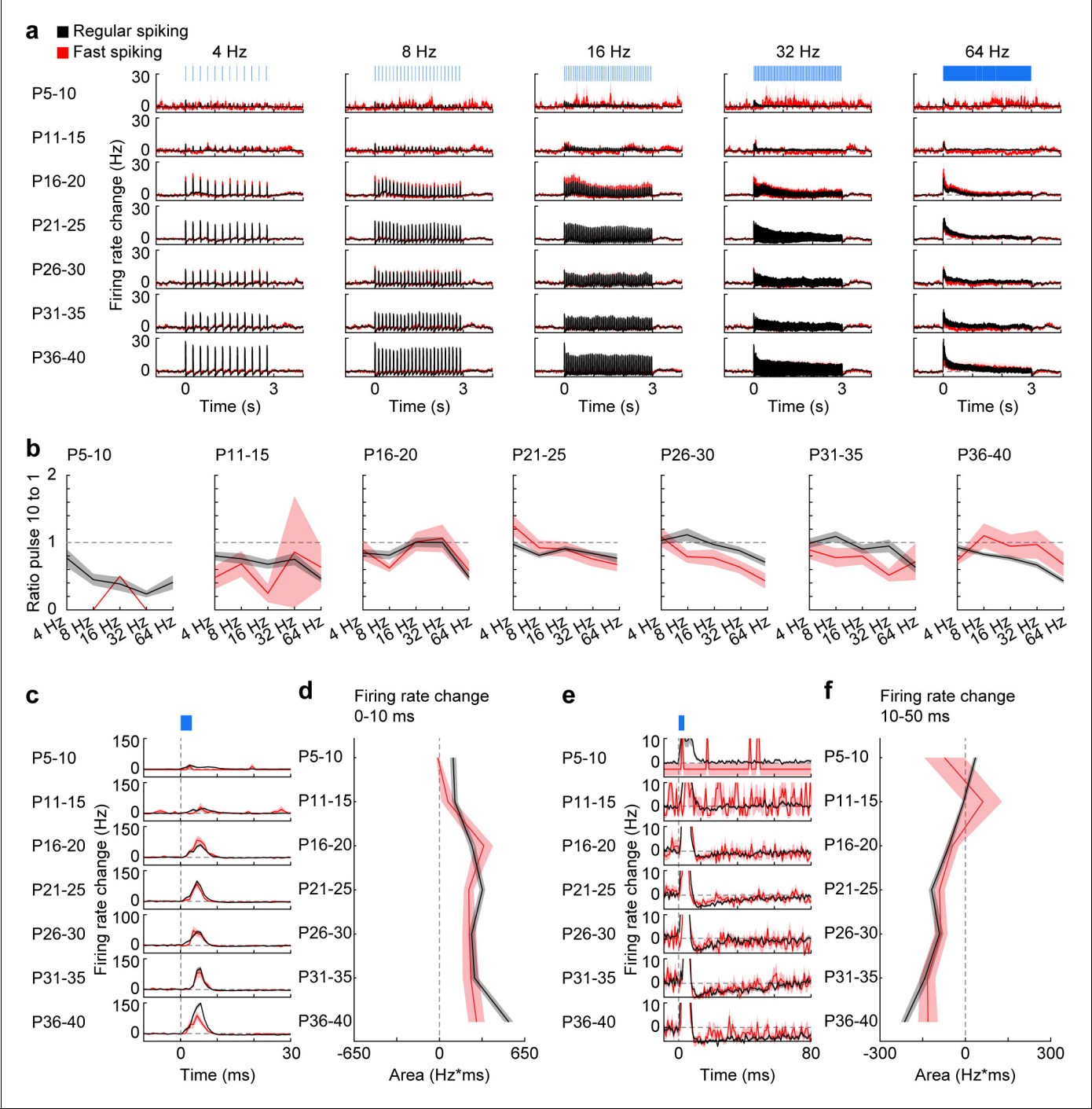

**Figure 6.** Firing of RS and FS units in response to pulse light stimulation. (a) Firing rate changes of prefrontal RS (black, n = 1824 units from 66 recordings/mice) and FS (red, n = 226 units from 66 recordings/mice) units in response to repetitive pulse light stimulation (3 ms, 473 nm) of 4, 8, 16, 32, and 64 Hz averaged for different age groups. (b) Line plots displaying the ratio of firing rate change in response to the 10[th] versus the 1[st] pulse for different frequencies and age groups (multifactorial ANOVA: unit type $F_{(1,7698)}$ = 0.39, p=0.530, age group $F_{(6, 7698)}$=18.7, p=1.00×10$^{-21}$, stimulation frequency $F_{(4, 7698)}$=36.0, p=6.29×10$^{-30}$). (c) Firing rate changes of RS and FS units in response to pulse light stimulation (3 ms, 473 nm) of L2/3 PYRs averaged for different age groups. (d) Line plot displaying the average firing rate change of RS and FS units 0–10 ms after pulse light stimulation for different age groups. (e) Same as (c) displayed at longer time scale. (f) Same as (d) for 10–50 ms after pulse start. (Average data is displayed as mean ± sem. See *Supplementary file 1* for a summary of experimental conditions. See *Supplementary file 2* for statistics).
The online version of this article includes the following source data and figure supplement(s) for figure 6:

**Source data 1.** Source data for *Figure 6b*.

*Figure 6 continued on next page*

Figure 6 continued

**Source data 2.** Source data for *Figure 6d*.
**Source data 3.** Source data for *Figure 6f*.
**Figure supplement 1.** Inter-spike intervals of RS and FS units during pulse light stimulation.

stimulation as best predictor variables (n = 48, df = 43, $R^2$ = 0.716, p=4.35×10$^{-12}$; see *Supplementary file 2* for detailed statistics).

Thus, peak amplitude during ramp light stimulation is correlated with firing rate changes in response to ramp and pulse stimulations, indicating the importance of stimulation efficacy for peak amplitude. On the other hand, FS characteristics and rhythmicity of single units show the strongest correlation with the peak frequency of LFP power.

## Discussion

Gamma oscillations (30–80 Hz) result from a fine-tuned interplay between excitation and inhibition in the adult brain (*Atallah and Scanziani, 2009*). For example, in sensory cortices, they arise through inhibitory feedback from PV-expressing interneurons (*Cardin et al., 2009*; *Chen et al., 2017*). Specifically, fast synaptic inhibition of the perisomatic region of pyramidal neurons by PV-expressing interneurons is critical for the generation of gamma oscillations (*Cardin, 2016*). Suppressing PV-expressing interneurons in the adult mPFC reduces the power in gamma frequency and impairs interhemispheric synchronization and cognitive abilities (*Cho et al., 2015*; *Cho et al., 2020*). SOM-expressing interneurons contribute to oscillatory activity at the lower end of the gamma frequency range (20–40 Hz) (*Chen et al., 2017*; *Veit et al., 2017*). In contrast, the mechanisms controlling the emergence of gamma activity during development are still poorly understood. Here, we reveal that the fast oscillatory activity in the mouse mPFC emerges during the second postnatal week and increases in frequency and amplitude before it stabilizes in gamma frequency range (30–80 Hz) during the fourth postnatal week. Further, we show that the functional maturation of FS PV-expressing interneurons and single unit rhythmicity of RS and FS units are best correlated with the accelerating gamma activity in the LFP. While activation of L2/3 PYR drives fast oscillatory activity throughout development, the acceleration toward higher frequencies relates to the maturation of inhibitory feedback and of FS interneurons. The time course of the maturation of FS units, putatively PV-expressing interneurons, is consistent with a recent characterization of PV-expressing interneuron physiology in the mPFC in vitro (*Miyamae et al., 2017*). These results suggest that the interplay between excitatory and inhibitory neurons is not only critical for the generation of adult gamma activity but also for its emergence during postnatal development.

The L2/3 PYRs-driven oscillatory activity was strongly correlated with spontaneously occurring fast frequency oscillations. However, broader peaks in power spectra for spontaneous activity indicate that the stimulation-induced activity might not cover the full diversity of spontaneous gamma activity. The broadness of spontaneous activity gamma power might also explain the absence of clear locking and rhythmicity of single units in the absence of prefrontal stimulation or task-induced activation of the mPFC.

Starting with the first electroencephalographic recordings, adult brain rhythms have been defined according to their frequencies and related to a specific state or task (*Buzsáki and Draguhn, 2004*). These 'classical' frequency bands (i.e. delta, theta, alpha, beta, gamma) are largely preserved between different mammalian species (*Buzsáki and Draguhn, 2004*; *Buzsáki et al., 2013*). However, how they emerge during development is still largely unknown. Synchronization of cortical areas in fast oscillatory rhythms starts during the first postnatal week (*Brockmann et al., 2011*). These low-amplitude patterns are detected in the rodent mPFC as early as P5,1–2 days later compared to primary sensory cortices (S1, V1) (*Minlebaev et al., 2011*; *Yang et al., 2013*; *Dupont et al., 2006*; *Shen and Colonnese, 2016*; *Yang et al., 2009*). However, this neonatal fast activity has a relative low frequency (around 20 Hz) (*Brockmann et al., 2011*; *Yang et al., 2009*). It is organized in infrequent short bursts and its detection is hampered by the lack of a clear peak in LFP power spectra. We previously showed that in the developing mPFC the detection of prominent fast oscillations with frequencies above 12 Hz coincides with the switch from discontinuous to continuous activity (*Brockmann et al., 2011*). These oscillations are initially within 15-20 Hz frequency range that was

classically defined as beta range. The present results indicate that these rhythms progressively increase their frequency and amplitude with age until they stabilize in gamma frequency range at 50–60 Hz during the fourth postnatal week. A similar increase in gamma frequency has been previously described after eye opening in the visual cortex (*Hoy and Niell, 2015*). Therefore, identification of oscillatory patterns in developing circuits according to 'classical' frequency bands established for adults should be avoided.

Adult gamma activity in the cerebral cortex relies on FS PV-expressing interneurons (*Cardin et al., 2009*). To test whether this mechanism underlies the fast rhythms in the developing brain, we developed an unbiased approach to detect FS units corresponding to putatively PV-expressing interneurons. Since PV expression and FS characteristics do not completely overlap (*Ma et al., 2006*; *Onorato et al., 2020*), the method has the same drawback as that typically used for the distinction of adult RS and FS and cannot identify RS interneurons. For this, clustering of prefrontal neurons from mice of all investigated ages was performed based on a dimensionality reduction of their mean waveforms and not on pre-defined waveform features. To validate this approach, we compared the results to pre-defined waveform features typically used to identify FS units and found that they largely agree for adult mice. We demonstrate that FS units are detected in the mPFC during the second postnatal week and progressively mature until the fourth postnatal week, consistent with PV interneuron maturation (*Okaty et al., 2009*). The similar dynamics of FS interneuron maturation and acceleration of fast oscillatory activity, as well as the correlation of FS characteristics with induced fast oscillation frequencies across age supports the hypothesis that FS interneurons are key elements for prefrontal gamma development. In the absence of FS interneurons at early age, inhibitory feedback from SOM neurons – important for slow gamma activity in the adult cortex (*Veit et al., 2017*) – might contribute to early oscillatory activity at frequencies within 12–20 Hz range.

While we only found minor age-dependent changes in the extracellular waveforms of RS units, an in-depth investigation of prefrontal PYRs during development had identified prominent changes in their dendritic arborization, passive and active membrane properties, as well as excitatory and inhibitory inputs (*Kroon et al., 2019*). These changes, even though not detected with extracellular recordings, most likely contribute to the maturation of pyramidal-interneuronal interactions and finally, of gamma activity. Indeed, we found that the maturation of inhibitory feedback in response to prefrontal L2/3 PYRs stimulation follows the same dynamics as the development of gamma oscillations. Furthermore, rhythmicity of single RS and FS units were identified as good predictors for the frequencies of oscillatory activity.

GABAergic transmission in the rodent cortex matures during postnatal development, reaching an adult-like state toward the end of the fourth postnatal week (*Le Magueresse and Monyer, 2013*; *Butt et al., 2017*; *Lim et al., 2018*). Shortly after birth, GABA acts depolarizing due to high intracellular chloride in immature neurons expressing low levels of the chloride cotransporter KCC2 relative to NKCC1 (*Rivera et al., 1999*; *Lim et al., 2018*). However, this depolarization is not sufficient to trigger action potential firing and results in shunting inhibition (*Kirmse et al., 2018*). The switch of GABA action from depolarizing to hyperpolarizing has been reported to occur during the second postnatal week (*Ben-Ari et al., 2012*), coinciding with the emergence of gamma band oscillations. Moreover, the composition of $GABA_A$-receptor subunits changes during postnatal development, causing a progressive decrease of decay-time constants of inhibitory postsynaptic currents (IPSCs) until they reach adult-like kinetics in the fourth postnatal week (*Okaty et al., 2009*; *Bosman et al., 2005*; *Laurie et al., 1992*). Simulations of neuronal networks proposed that increasing IPSCs kinetics in FS interneurons results in increasing gamma frequency (*Doischer et al., 2008*). The gradual increase of prefrontal gamma frequency from the second to the fourth postnatal week provides experimental evidence for this hypothesis.

In-depth understanding of the dynamics and mechanisms of gamma activity in the developing cortex appears relevant for neurodevelopmental disorders, such as schizophrenia and autism. Both, in patients and disease mouse models, gamma oscillations have been reported to be altered, likely to reflect abnormal pyramidal-interneuronal interactions (*Cho et al., 2015*; *Cao et al., 2018*; *Rojas and Wilson, 2014*). These dysfunction seems to emerge already during development (*Chini et al., 2020*; *Richter et al., 2019*; *Hartung et al., 2016*). Elucidating the developmental dynamics of cortical gamma activity might uncover the timeline of disease-related deficits.

# Materials and methods

## Key resources table

| Reagent type (species) or resource | Designation | Source or reference | Identifiers | Additional information |
|---|---|---|---|---|
| Antibody | Rabbit polyclonal-anti-parvalbumin | Abcam | ab11427 | (1:500) |
| Antibody | Rabbit polyclonal-anti-somatostatin | Santa Cruz | sc13099 | (1:250) |
| Antibody | Goat-anti-rabbit secondary antibody, Alexa Fluor 488 | Invitrogen-Thermo Fisher | A11008 | (1:500) |
| Chemical compound, drug | Isoflurane | Abbott | B506 | |
| Chemical compound, drug | Urethane | Fluka analytical | 94300 | |
| Strain, strain background (mouse, both genders) | C57Bl/6J | Universitätsklinikum Hamburg-Eppendorf –Animal facility | C57Bl/6J | https://www.jax.org/strain/008199 |
| Recombinant DNA reagent | pAAV-CAG-ChR2(E123T/T159C)—2AtDimer2 | Provided by T. G. Oertner | pAAV-CAG-ChR2(E123T/T159C)—2AtDimer2 | http://www.oertner.com/ |
| Software, algorithm | Matlab R2018b | MathWorks | Matlab R2018b | https://www.mathworks.com/ |
| Software, algorithm | Kilosort2 | MouseLand | | https://github.com/MouseLand/Kilosort2 |
| Software, algorithm | ImageJ | ImageJ | | https://imagej.nih.gov/ij/ |
| Other | Arduino Uno SMD | Arduino | A000073 | |
| Other | Digital Lynx 4SX | Neuralynx | Digital Lynx 4SX | http://neuralynx.com/ |
| Other | Diode laser (473 nm) | Omicron | LuxX 473–100 | |
| Other | Electroporation device | BEX | CUY21EX | |
| Other | Electroporation tweezer-type paddles | Protech | CUY650-P5 | |
| Other | Recording electrode (one-shank, 16 channels) | Neuronexus | A1 × 16 5 mm | |
| Other | Recording electrode (four-shank, 16 channels) | Neuronexus | A4 × 4 5 mm | |

## Animals

All experiments were performed in compliance with the German laws and the guidelines of the European Community for the use of animals in research and were approved by the local ethical committee (G132/12, G17/015, N18/015). Timed-pregnant mice from the animal facility of the University Medical Center Hamburg-Eppendorf were housed individually at a 12 hr light/12 hr dark cycle and were given access to water and food ad libitum. The day of vaginal plug detection was considered embryonic day (E) 0.5, the day of birth was considered postnatal day (P) 0. Experiments were carried out on C57Bl/6J mice of both sexes.

## In utero electroporation (IUE)

Pregnant mice (C57Bl6/J, The Jackson Laboratory, ME, USA) received additional wet food daily, supplemented with 2–4 drops Metacam (0.5 mg/ml, Boehringer-Ingelheim, Germany) one day before until two days after in IUE. At E15.5, pregnant mice were injected subcutaneously with buprenorphine (0.05 mg/kg body weight) 30 min before surgery. Surgery was performed under isoflurane anesthesia (induction 5%, maintenance 3.5%) on a heating blanket. Eyes were covered with eye ointment and pain reflexes and breathing were monitored to assess anesthesia depth. Uterine horns were exposed and moistened with warm sterile PBS. 0.75–1.25 µl of opsin- and fluorophore-encoding plasmid (pAAV-CAG-ChR2(E123T/T159C)−2A-tDimer2, 1.25 µg/µl) purified with NucleoBond (Macherey-Nagel, Germany) in sterile PBS with 0.1% fast green dye was injected in the right lateral ventricle of each embryo using pulled borosilicate glass capillaries. Electroporation tweezer paddles of 5 mm diameter were oriented at a rough 20° leftward angle from the midline of the head and a rough 10° downward angle from the anterior to posterior axis to transfect precursor cells of medial prefrontal layer 2/3 PYRs neurons with five electroporation pulses (35 V, 50 ms, 950 ms interval, CU21EX, BEX, Japan). Uterine horns were placed back into the abdominal cavity. Abdominal cavity was filled with warm sterile PBS and abdominal muscles and skin were sutured with absorbable and non-absorbable suture thread, respectively. After recovery from anesthesia, mice were returned to their home cage, placed half on a heating blanket for two days after surgery. Fluorophore expression was assessed at P2 in the pups with a portable fluorescence flashlight (Nightsea, MA, USA) through the intact skin and skull and confirmed in brain slices postmortem.

## Electrophysiology
### Acute head-fixed recordings

Multi-site extracellular recordings were performed unilaterally or bilaterally in the mPFC of non-anesthetized or anesthetized P5-40 mice. Mice were on a heating blanket during the entire procedure. Under isoflurane anesthesia (induction: 5%; maintenance: 2.5%), a craniotomy was performed above the mPFC (0.5 mm anterior to bregma, 0.1–0.5 mm lateral to the midline). Pups were head-fixed into a stereotaxic apparatus using two plastic bars mounted on the nasal and occipital bones with dental cement. Multi-site electrodes (NeuroNexus, MI, USA) were inserted into the mPFC (four-shank, A4 × 4 recording sites, 100 µm spacing, 125 µm shank distance, 1.8–2.0 mm deep). A silver wire was inserted into the cerebellum and served as ground and reference. For non-anesthetized and anesthetized recordings, pups were allowed to recover for 30 min prior to recordings. For anesthetized recordings, urethane (1 mg/g body weight) was injected intraperitoneally prior to the surgery.

## Acute head-fixed recordings with chronically implanted head-fixation adapters

Multisite extracellular recordings were performed unilaterally in the mPFC of P23-25 and P38-40 mice. The adapter for head fixation was implanted at least 5 days before recordings. Under isoflurane anesthesia (5% induction, 2.5% maintenance), a metal head-post (Luigs and Neumann, Germany) was attached to the skull with dental cement and a craniotomy was performed above the mPFC (0.5–2.0 mm anterior to bregma, 0.1–0.5 mm right to the midline) and protected by a customized synthetic window. A silver wire was implanted between skull and brain tissue above the cerebellum and served as ground and reference. 0.5% bupivacaine/1% lidocaine was locally applied to cutting edges. After recovery from anesthesia, mice were returned to their home cage. After recovery from the surgery, mice were accustomed to head-fixation and trained to run on a custom-made spinning

disc. For non-anesthetized recordings, craniotomies were uncovered and multi-site electrodes (NeuroNexus, MI, USA) were inserted into the mPFC (one-shank, A1 × 16 recording sites, 100 µm spacing, 2.0 mm deep).

Extracellular signals were band-pass filtered (0.1–9000 Hz) and digitized (32 kHz) with a multi-channel extracellular amplifier (Digital Lynx SX; Neuralynx, Bozeman, MO, USA). Electrode position was confirmed in brain slices postmortem.

### Optogenetic stimulation

Ramp (linearly increasing light power) light stimulation and pulsed (short pulses of 3 ms) light stimulation at different frequencies was performed using an Arduino uno (Arduino, Italy) controlled laser system (473 nm / 594 nm wavelength, Omicron, Austria) coupled to a 50 µm (four- shank electrodes) or 105 µm (one-shank electrodes) diameter light fiber (Thorlabs, NJ, USA) glued to the multisite electrodes, ending 200 µm above the top recording site. Each type of stimulation was repeated 30 times. At the beginning of each recording, laser power was adjusted to reliably trigger neuronal spiking in response to light pulses of 3 ms duration.

### Histology

Mice (P5-40) were anesthetized with 10% ketamine (aniMedica, Germanry)/2% xylazine (WDT, Germany) in 0.9% NaCl (10 µg/g body weight, intraperitoneal) and transcardially perfused with 4% paraformaldehyde (Histofix, Carl Roth, Germany). Brains were removed and postfixed in 4% paraformaldehyde for 24 hr. Brains were sectioned coronally with a vibratom at 50 µm for immunohistochemistry.

### Immunohistochemistry

Free-floating slices were permeabilized and blocked with PBS containing 0.8% Triton X-100 (Sigma-Aldrich, MO, USA), 5% normal bovine serum (Jackson Immuno Research, PA, USA) and 0.05% sodium azide. Slices were incubated over night with primary antibody rabbit-anti-parvalbumin (1:500, #ab11427, Abcam, UK) or rabbit-anti-somatostatin (1:250, #sc13099, Santa Cruz, CA, USA), followed by 2 hr incubation with secondary antibody goat-anti-rabbit Alexa Fluor 488 (1:500, #A11008, Invitrogen-Thermo Fisher, MA, USA). Sections were transferred to glass slides and covered with Fluoromount (Sigma-Aldrich, MO, USA).

### Cell quantification

Images of immunofluorescence in the right mPFC were acquired with a confocal microscope (DM IRBE, Leica, Germany) using a 10x objective (numerical aperture 0.3). Immunopositive cells were automatically quantified with custom-written algorithms in ImageJ environment. The region of interest (ROI) was manually defined over L2/3 of the mPFC. Image contrast was enhanced before applying a median filter. Local background was subtracted to reduce background noise and images were binarized and segmented using the watershed function. Counting was done after detecting the neurons with the extended maxima function of the MorphoLibJ plugin.

### Data analysis

Electrophysiological data were analyzed with custom-written algorithms in Matlab environment. Data were band-pass filtered (500–9000 Hz for spike analysis or 1–100 Hz for local field potentials (LFP)) using a third-order Butterworth filter forward and backward to preserve phase information before down-sampling to analyze LFP. Each type of optogenetic stimulation (ramps or pulses at different frequencies) was repeated 30 times for each recording. Each recording contributes a single data point for *Figures 1* and *3*. Most mice were recorded once. Only mice with chronically implanted head fixation were recorded more than one time (13 recordings from six mice at P24-26 and 12 recordings from five mice at P37-40), but all recordings were with acutely inserted electrodes. All mice for single unit analysis were recorded only once. (See *Supplementary file 1* for a summary of experimental conditions.)

## Power spectral density

For power spectral density analysis, 2 s-long windows of LFP signal were concatenated and the power was calculated using Welch's method with non-overlapping windows. Spectra were multiplied with squared frequency.

## Modulation index

For optogenetic stimulations, MI was calculated as (value stimulation - value pre- stimulation) / (value stimulation + value pre-stimulation).

## Peak frequency and strength

Peak frequency and peak strength were calculated for the most prominent peak in the spectrum defined by the product of peak amplitude, peak half width and peak prominence.

## Single unit analysis

Spikes were detected and sorted with Kilosort2 in Matlab. t-sne dimensionality reduction was applied on mean waveforms of all units. Hierarchical clustering was performed to identify FS and RS units for all ages simultaneously. Autocorrelations of single units with a minimum firing rate of 1 Hz were calculated before and during optogenetic ramp stimulation. Power spectral densities of mean autocorrelations were calculated per unit. Crosscorrelations of simultaneously recorded RS-RS, FS-FS, and RS-FS unit pairs were calculated during optogenetic ramp stimulation. Power of the mean spike-triggered LFP, pairwise phase consistency (*Vinck et al., 2010*), coefficient of variation, $C_{V2}$ (*Holt et al., 1996*), and inter-spike intervals were calculated for individual units before and during optogenetic ramp stimulation.

## Statistics

No statistical measures were used to estimate sample size since effect size was unknown. Data were tested for consistent trends across age with the non-parametric Mann-Kendall trend test. Mann-Kendall coefficient tau-b adjusting for ties is reported. Multifactorial ANOVAs were used to compare main effects. Linear regression models were used to test for significant links of averaged single unit and LFP activity measures with age per recording. Stepwise linear regression was used to identify significant links from averaged single unit activity measures (proportion FS units, as well as half width, trough to peak, amplitude, rate change for ramp light stimulation, autocorrelation peak during ramp light stimulation, spike-triggered LFP peak frequency during ramp stimulation, and rate change early (0–10 ms) and late (10–50 ms) after pulse stimulation for RS and FS units) with LFP activity (ramp light-induced frequency peak and amplitude). See *Supplementary file 2* for detailed statistics.

## Acknowledgements

We thank M Chini for helpful discussions and comments on the manuscript as well as A Marquardt, P Putthoff, A Dahlmann, and K Titze for excellent technical assistance. This work was funded by grants from the European Research Council (ERC-2015-CoG 681577 to ILH-O) and the German Research Foundation (Ha 4466/10–1, Ha4466/11-1, Ha4466/12-1, SPP 1665, SFB 936 B5 to ILH-O).

## Additional information

### Funding

| Funder | Grant reference number | Author |
| --- | --- | --- |
| H2020 European Research Council | ERC-2015-CoG 681577 | Ileana Hanganu-Opatz |
| Deutsche Forschungsge-meinschaft | Ha4466/10-1 | Ileana Hanganu-Opatz |
| Deutsche Forschungsge-meinschaft | Ha4466/11-1 | Ileana Hanganu-Opatz |

| Deutsche Forschungsge-meinschaft | Ha4466/12-1 | Ileana Hanganu-Opatz |
| Deutsche Forschungsge-meinschaft | SPP 1665 | Ileana Hanganu-Opatz |
| Deutsche Forschungsge-meinschaft | SFB 936 B5 | Ileana Hanganu-Opatz |

The funders had no role in study design, data collection and interpretation, or the decision to submit the work for publication.

## Author contributions

Sebastian H Bitzenhofer, Conceptualization, Data curation, Software, Formal analysis, Supervision, Investigation, Visualization, Methodology, Writing - original draft, Writing - review and editing; Jastyn A Pöpplau, Data curation, Formal analysis, Investigation, Methodology, Writing - review and editing; Ileana Hanganu-Opatz, Conceptualization, Supervision, Funding acquisition, Project administration, Writing - review and editing

## Author ORCIDs

Sebastian H Bitzenhofer (iD) https://orcid.org/0000-0003-0736-6251
Jastyn A Pöpplau (iD) https://orcid.org/0000-0002-4350-3164
Ileana Hanganu-Opatz (iD) https://orcid.org/0000-0002-4787-1765

## Ethics

Animal experimentation: All experiments were performed in compliance with the German laws and the guidelines of the European Community for the use of animals in research and were approved by the local ethical committee (G132/12, G17/015, N18/015).

## Decision letter and Author response

Decision letter https://doi.org/10.7554/eLife.56795.sa1
Author response https://doi.org/10.7554/eLife.56795.sa2

# Additional files

## Supplementary files

• Supplementary file 1. Recording summary. Table summarizing the recordings for each experimental condition.

• Supplementary file 2. Detailed statistical results. Table summarizing the statistical results.

• Transparent reporting form

## Data availability

The authors declare that all data and code supporting the findings of this study are included in the manuscript. LFP and SUA data for all recordings is available at the following open-access repository: https://doi.org/10.12751/g-node.heyl6r.

The following dataset was generated:

| Author(s) | Year | Dataset title | Dataset URL | Database and Identifier |
|---|---|---|---|---|
| Bitzenhofer SB, Pöpplau A, Hanganu-Opatz IL | 2020 | Ephys data associated with the paper *Gamma activity accelerates during prefrontal development* | https://doi.org/10.12751/g-node.heyl6r | g-node, 10.12751/g-node.heyl6r |

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
