## [Decision Letter]

**Acceptance summary:**

Bitzenhofer et al., show that fast network oscillations in the γ-frequency range become progressively stronger during early development, and stabilize around the fourth postnatal week. This development shows a similar temporal progression and is correlated with the maturation of fast spiking interneurons, a cell class that is known to regulate fast network oscillations. These findings provide important new insights into the development of γ oscillations and will be valuable for understanding the development of neuropsychiatric diseases linked to GABAergic dysfunction and abnormal oscillatory activity.

**Decision letter after peer review:**

Thank you for submitting your article "Γ activity accelerates during prefrontal development" for consideration by *eLife*. Your article has been reviewed by four peer reviewers, one of whom is a member of our Board of Reviewing Editors, and the evaluation has been overseen by Laura Colgin as the Senior Editor. The following individuals involved in review of your submission have agreed to reveal their identity: Quentin Perrenoud (Reviewer #3); Christopher I Moore (Reviewer #4).

The reviewers have discussed the reviews with one another and the Reviewing Editor has drafted this decision to help you prepare a revised submission.

Summary:

Bitzenhofer et al., examined the developmental trajectory of γ-band oscillations during early development in mice prefrontal cortex. The authors used extracellular recordings and optogenetic stimulations in mice aged P5-40. Rhythmic activity in the prefrontal cortex became more prominent during the second postnatal week and increased in frequency. Developmental modifications in the strength of γ-band oscillations correlated with activation of layer 2/3 pyramidal neurons while the acceleration of their frequency followed similar temporal dynamics as the maturation of fast-spiking interneurons. The current findings provide important data on the maturation of rhythmic activity and the underlying mechanisms in prefrontal circuits. Given the relative scarcity of data in this field, the findings are potentially important and of general interest.

The reviewers made many positive comments on your manuscript. They think the study fills an important gap in terms of developmental studies on oscillations and provides mechanistic insight into the emergence of γ rhythms. They agree that the dataset is rich and contains many challenging experiments providing valuable data that is scarce in the field. They also consider that the findings fit well with the existing literature.

Essential revisions:

The reviewers were generally appreciative of the quality and richness of the experiments. Their main concerns were the analysis and interpretation of the data. The reviewers assess that this work can be done without further experiments, but by further analysis.

1) The greatest concern pertains to the claims about the mechanisms underlying the development of rhythms and their statistical support.

This refers to claims like, "These results demonstrate that the interplay between excitatory and inhibitory neurons is not only critical for the generation of adult γ activity but also for its emergence during postnatal development."

A core argument is that developmental modifications in high-frequency activity mirror the changes observed in both PV/FS-activation and L2/3 pyramidal neurons. The authors refer frequently to the fact the patterns of electrophysiological activity "mirror" those observed, for example, for the maturation of FS interneurons. The authors should investigate these relationships more formally, rather stating that there are similar trends in the data, for example, through analysis of individual changes.

Overall, the authors make little effort to test this link statistically and their claim is only backed by observational and potentially subjective similarities between developmental trends.

Specifically, reviewers commented that all the data presented in the manuscript, except for the immunohistochemistry, come from the same data-set. It should thus be possible to test whether a significant link exists across recordings between the power or the mode of high frequency activity and the various metrics used by the authors to quantify RS and FS firing. A variety of methods exist for this purpose and would be acceptable here (a stepwise multi-linear regression for instance).

The authors should also characterize the development of synchrony and phase-locking over time. How does synchrony amongst FS evolve, for example, over time?

In general, the study makes strong claims like "demonstrate that the interplay between excitatory and inhibitory neurons is not only critical for the generation of adult γ activity but also for its emergence during postnatal development"(Discussion section). Most of the observation provided by the author are based on correlations. Thus, we encourage the author to be more cautious in their conclusions.

2) Important analyses about neural activity are missing that are critical for the interpretation of the data. Overall, the analysis of the spiking and rhythmicity of RS and FS unit is somewhat limited. The reviewers feel that adding these analyses are necessary to make the paper more complete and impactful.

Requests:

i) Pairwise analyses of synchrony amongst FS over time. Important from many perspectives, including relating these data to the slice literature where younger animals are often used e.g. to examine coupling frequency.

ii) The authors should see whether other spiking metrics evolve over time: CV2 is one among many, ISI probability (as distinct from frequency analysis of the spike train--these do give a different perspective).

iii) The manuscript does not provide of direct measure of the phase-locking of RS and FS units to high-frequency activity. The sole metric used for this purpose is the autocorrelation which is rather a measure of intrinsic rhythmicity than a measure of the entrainment to LFP signals. It is also important to note that, while γ activity is often conceived as a sustained oscillation, studies have shown that in most cortical regions it patterns in short bouts (Siegle et al., 2014) having dynamics that are similar to filtered white noise (Burns et al., 2011). Thus, an increase in γ activity is generally not accompanied by an increase of rhythmicity in the autocorrelation of RS and FS units (Perrenoud et al., 2016). A more appropriate approach would be to estimate the phase of high-frequency activity using a wavelet or a Hilbert transform (Bruns, 2004) and to quantify the phase locking of RS and FS units using either spike-field coherence, or if firing bias is a concern, with the Pairwise Phase Consistency.

iv) The authors should do a more sensitive quantification of spike-field coherence for spontaneous firing, because autocorrelations will be highly insensitive when the firing rates are low. Otherwise, what does the γ in the PFC reflect, if not the firing of individual neurons?

v) Given the richness of the dataset and the policies of *eLife* with respect to data publication, we strongly encourage the authors to published their raw data which may allow other aspects of this dataset to be analysed in the future.

3) Spectral analyses. The reviewers requested a more refined spectral analysis:

i) The authors examine spontaneous activity (Figure 1) and compare these data to optogenetically driven oscillations (Figure 3). However, inspection of spectrograms (Figure 1 and Figure 3B) may reveal some differences. The spontaneous high-frequency activity appears much more broadbanded (Figure 1B) than the optogenetically driven activity (Figure 3B). This raises the question whether the patterns are actually reflecting the same process and thus potentially involving different mechanisms (broad-band modulation vs. narrow-banded oscillation). Similarly, changes in peak-frequency vs. increases in power may reflect quite distinct phenomena that should be disentangled. In this context, did the authors examine also changes in lower frequency power across development? Close inspection of Figure 1B reveals a clear peak in the δ/theta-range at P40 which is not present in other groups.

ii) Reviewers commented that there is very little low frequency power for the P5-10 group. Reviewers would like to see what happens if the authors normalize the power differently (using a standard division by the total power), and if there would not be more γ power at high frequencies?

Related to this, reviewers commented that the spontaneous firing rates seem to be very low for the P5-10 group (see Figure 4) and were concerns that comparing absolute power across animals amounts to comparing mean firing rates.

iii) The lower peak seems to shift to a lower frequency with age, rather than to a higher frequency. The authors should discuss this.

iv) There appear to be artifacts in the spectrum shown in Figure 1, column 3, second row P12 – what are these artifacts? Artifacts are peaks separated by about 10Hz, meaning they are 0.1s artifacts, this could be due to anesthesia apparatus? Could this have influenced the quantification of γ?

v) Figure 3C: From this figure, the developmental change in spectral power is not clearly visible. The authors may want to use a different way of highlighting changes in spectral power across development.

4) Effect of behavioural state. The authors should address the issue of behavioural state.

In general comparisons are made between animals where it is not clear that the behavioural state is the same. For example, chronic recordings were done only in P25 and P40 groups, but the behavioural state of the younger (and the conditions of recordings) during wakefulness are completely unclear. It is not clear whether changes in γ observed might be due to behavioural state (see also Hoy and Niell, 2015). Was the behavioral state the same between mice? Did mice at different ages run more than at other ages? How did running impact FS or RS firing?

The reviewers further wondered how the time in the session impacted likelihood of seeing different rhythms (did lower frequencies predominate later in the session, as many prior studies would predict)?

Overall, the authors feel that the data in this sense is underused and that there may be a lot of useful developmental richness that would make the paper a more important contribution.

The Materials and methods section says some wake animals were recorded from a stereotaxis apparatus. What does this apparatus imply for behavioral state?

The conditions of the recordings need to be clearly described.

5) Richer analyses of the adaptation functions is highly warranted, given interesting work (e.g., by Llampl and many others) in the adaptation of inhibitory versus excitatory currents.

Reviewers further wondered if the difference in apparent adaptation cannot be simply explained by the overall difference in activation to light trains?

6) While FS units are generally assumed to correspond to PV interneurons, pyramidal neurons (Onorato et al., 2019) and Somatostatin interneurons (Ma et al., 2006) can also display these characteristics. Thus, the FS units identified in this study might not only include PV interneurons. The authors should discuss this.

The authors wondered if the authors can give a strong argument or piece data that the "fast-spiking firing properties" are also altered, in the sense of the neurons' input-output functions (how they respond to current or synaptic input). That is fast spiking needs to be more clearly defined and interpreted.

7) In general reviewers agreed that the dataset (nature of the statistical units, mice, recording and state – awake, anesthetized, sex) is not described clearly enough. These points would greatly improve the paper.

i) Reviewers noted it is generally difficult to figure out how the authors exactly analyzed their data, what data points went in the figures, etc. Tables summarizing the dataset would be very useful.

ii) Chronic recordings were done only in P23-25 and P38-40 – are these difference mice?

Figure 1C – are these different mice? If so, can we compare them to each other? What data goes exactly in this figure in terms of acute recordings and chronic recordings for 1A-C. It seems for the wake group the authors compare chronic to non-chronic conditions. That comparison seems problematic given the effect on behavioural state.

iii) N = 114 recordings, but n=80 and n=20,35 in the figure legends what does it refer to? The n's used for the analysis and the number of mice should be clear everywhere in the paper. Is every mouse contributing a separate data point for the statistics, or is every channel, or every recording a data point? If so, isn't the analysis problematic because you are inflating your statistical degrees of freedom massively. For example, p=2.73*10-8 seems to be a p-value that is suffering from inflation due to dependent measures. Similar comments apply to the rest of the paper.

It is unclear how sample size impact the inferences we should take from different plots and analyses--many more cells are sampled at certain days. For example, in Figure 6B, there seem to be differences in the adaptation functions, but it's not clear whether this is the case.

iv) Figure 1B panel right -there are no error bars on this figure, how reliable are these differences across mice? Figure 1 Heatmap – it seems that there is a lot variability, perhaps across different mice? The authors should show how consistent the effects are across mice that are recorded under similar conditions.

v) The authors should describe the intensity of the laser, and say how the intensity was calibrated and determined. The reviewers wondered whether the laser is actually driving activity in the pups, because this is not clearly visible from the plots, perhaps due to extremely low spontaneous firing rates. It appears that high frequency activity (which is dominated by spiking) does not seem to be affected by the laser. Single examples of neurons with raster plots would be very helpful to understand this. In general, providing raster plots with single spikes would be extremely helpful for the reader to assess the reliability of optogenetics simulation and the meaning of various statistics computed. The authors should ideally show an example of a single neuron that is positively modulated by light stimulation for the young pups to understand what is going on.

The text should clearly distinguish between absolute rate changes and modulation. If spontaneous firing rates are very low, then neurons might show a large relative change, but they could still be largely unresponsive to the light.

vi) The following statement is not clear. The first p-value doesn't seem to be significant.

"While ramp-induced firing rate changes of RS units (Mann207 Kendall trend test, p=0.07, n=7 age groups, tau-b 0.619) became more prominent at older 208 age, the firing rate changes were stable for FS units (Mann-Kendall trend test, p=0.88, 209 n=7 age groups, tau-b 0.047) (Figure 4B)"

vii) "whereas at older age the number 211 of activated and inactivated RS units got more balanced (Mann-Kendall trend test, p=1.52*10-14 212 , n=1821 units, tau-b -0.123)"

Please explain what "more balanced" means and how balanced is quantified.

viii) Reviewers commented that in Figure 5, it is unclear what the quantification or statistical test is. It looks like only the P36-40 is qualitatively different, and the P5-10. But the intermediate ages do not seem to show a consistent pattern.

(viiii) A variety of methods are available for hierarchical clustering. However, a reviewer did not understand which of these the authors use for the clustering of single units. Please make sure it is well detailed in the Materials and methods sections.

Comments on literature:

i) The authors highlight in this paper the importance of early development for the maturation of neural oscillations. It would be useful to expand the literature review to indicate that development of rhythmic activity as well as the underlying mechanism extend beyond P40.

ii) In the Introduction, please specify that the results obtained by Chen et al., 2017; Veit et al., 2017 were acquired in the visual cortex as the type of β activity observed in these studies has not yet been observed in other regions.

iii) Discussion and comparison with existing literature (Hoy and Niell, 2015 – which was not cited) is necessary. That study contains several findings reported here as well.

---

## [Author Response]

Essential revisions:The reviewers were generally appreciative of the quality and richness of the experiments. Their main concerns were the analysis and interpretation of the data. The reviewers assess that this work can be done without further experiments, but by further analysis.

We thank the reviewers for the constructive feedback and most helpful comments.

1) The greatest concern pertains to the claims about the mechanisms underlying the development of rhythms and their statistical support.This refers to claims like, "These results demonstrate that the interplay between excitatory and inhibitory neurons is not only critical for the generation of adult γ activity but also for its emergence during postnatal development."

We rephrased and toned down our statements.

A core argument is that developmental modifications in high-frequency activity mirror the changes observed in both PV/FS-activation and L2/3 pyramidal neurons. The authors refer frequently to the fact the patterns of electrophysiological activity "mirror" those observed, for example, for the maturation of FS interneurons. The authors should investigate these relationships more formally, rather stating that there are similar trends in the data, for example, through analysis of individual changes.

We used linear regression models to test for significant links between average measures of individual recordings and age and added the results to the Results section and Supplementary file 2. Further, we performed stepwise linear regressions to identify significantly links between these measures and oscillatory activity in the local field potential (LFP) (see below).

Overall, the authors make little effort to test this link statistically and their claim is only backed by observational and potentially subjective similarities between developmental trends.Specifically, reviewers commented that all the data presented in the manuscript, except for the immunohistochemistry, come from the same data-set. It should thus be possible to test whether a significant link exists across recordings between the power or the mode of high frequency activity and the various metrics used by the authors to quantify RS and FS firing. A variety of methods exist for this purpose and would be acceptable here (a stepwise multi-linear regression for instance).

We thank the reviewers for the suggestion. We performed stepwise linear regressions on average values for individual mice to examine links between single unit measures and oscillatory activity in the LFP. We added the results to the Results section and to Supplementary file 2.

The authors should also characterize the development of synchrony and phase-locking over time. How does synchrony amongst FS evolve, for example, over time?

In line with the suggestion, we further analyzed the development of synchrony using cross-correlations of simultaneously recorded FS-FS, RS-RS, and RS-FS unit pairs and added the results as Figure 5—figure supplement 3. We analyzed phase locking using pairwise phase consistency and power of average spike-triggered LFP for spontaneous activity and during ramp light stimulation and added the results to Figure 5, Figure 5—figure supplement 1, and Figure 5—figure supplement 2.

In general, the study makes strong claims like "demonstrate that the interplay between excitatory and inhibitory neurons is not only critical for the generation of adult γ activity but also for its emergence during postnatal development"(Discussion section). Most of the observation provided by the author are based on correlations. Thus, we encourage the author to be more cautious in their conclusions.

We rephrased and toned down the statements.

2) Important analyses about neural activity are missing that are critical for the interpretation of the data. Overall, the analysis of the spiking and rhythmicity of RS and FS unit is somewhat limited. The reviewers feel that adding these analyses are necessary to make the paper more complete and impactful.Requests:i) Pairwise analyses of synchrony amongst FS over time. Important from many perspectives, including relating these data to the slice literature where younger animals are often used e.g. to examine coupling frequency.

We analyzed cross-correlations of simultaneously recorded RS-RS, FS-FS, and RS-FS unit pairs during ramp light stimulation across age. We added the new results to the manuscript (Discussion section) and as Figure 5—figure supplement 3. The additional analyses showed no rhythmic interactions of RS-RS unit pairs at young age, but at older age with increasing strength and frequency. Similar results were obtained for FS-FS and RS-FS unit pairs, but data were less clear due to the low numbers of simultaneously recorded FS unit pairs. No consistent time lag was detected for RSFS unit pairs interactions.

ii) The authors should see whether other spiking metrics evolve over time: CV2 is one among many, ISI probability (as distinct from frequency analysis of the spike train--these do give a different perspective).

In line with the suggestion, we complemented the rhythmicity analysis using autocorrelations with two other measures: (i) the coefficient of variation (CV) of inter-spike intervals, C_V2_ and (ii) interspike intervals for RS and FS units. We added the new results to the manuscript (subsection “The rhythmicity of pyramidal cell and interneuron firing follows similar development as accelerating γ activity in mPFC”) and displayed the data as Figure 5—figure supplement 2. We show that RS units of older mice have longer inter-spike intervals (at lower instantaneous frequency), indicating that they have a higher tendency to skip cycles of fast oscillatory activity.

iii) The manuscript does not provide of direct measure of the phase-locking of RS and FS units to high-frequency activity. The sole metric used for this purpose is the autocorrelation which is rather a measure of intrinsic rhythmicity than a measure of the entrainment to LFP signals. It is also important to note that, while γ activity is often conceived as a sustained oscillation, studies have shown that in most cortical regions it patterns in short bouts (Siegle et al., 2014) having dynamics that are similar to filtered white noise (Burns et al., 2011). Thus, an increase in γ activity is generally not accompanied by an increase of rhythmicity in the autocorrelation of RS and FS units (Perrenoud et al., 2016). A more appropriate approach would be to estimate the phase of high-frequency activity using a wavelet or a Hilbert transform (Bruns, 2004) and to quantify the phase locking of RS and FS units using either spike-field coherence, or if firing bias is a concern, with the Pairwise Phase Consistency.

In line with the suggestion, we analyzed the power of average spike-triggered LFP, as well as pairwise phase consistency for RS and FS units during spontaneous activity and ramp light stimulations across age. We added the new results to the manuscript (subsection “The rhythmicity of pyramidal cell and interneuron firing follows similar development as accelerating γ activity in mPFC”) and displayed them in Figure 5—figure supplement 1 and Figure 5—figure supplement 2. These novel findings are consistent with the idea that γ activity in the LFP reflects rhythmic activity of single units in the medial prefrontal cortex (mPFC).

iv) The authors should do a more sensitive quantification of spike-field coherence for spontaneous firing, because autocorrelations will be highly insensitive when the firing rates are low. Otherwise, what does the γ in the PFC reflect, if not the firing of individual neurons?

As recommended, we added additional measures to quantify spike-field coherence (see point 2 (iii)). Autocorrelations, pairwise phase consistency, and power of average spike-triggered LFP did not show obvious rhythmicity during spontaneous activity. We assume this might be due to the short and low power γ activity in the absence of a task engaging the mPFC. We added a short discussion of this topic to the text (Discussion section).

v) Given the richness of the dataset and the policies of eLife with respect to data publication, we strongly encourage the authors to published their raw data which may allow other aspects of this dataset to be analysed in the future.

We made the dataset available online. https://doi.org/10.12751/g-node.heyl6r

3) Spectral analyses. The reviewers requested a more refined spectral analysis:i) The authors examine spontaneous activity (Figure 1) and compare these data to optogenetically driven oscillations (Figure 3). However, inspection of spectrograms (Figure 1 and Figure 3B) may reveal some differences. The spontaneous high-frequency activity appears much more broadbanded (1B) than the optogenetically driven activity (3b). This raises the question whether the patterns are actually reflecting the same process and thus potentially involving different mechanisms (broad-band modulation vs. narrow-banded oscillation). Similarly, changes in peak-frequency vs. increases in power may reflect quite distinct phenomena that should be disentangled. In this context, did the authors examine also changes in lower frequency power across development? Close inspection of Figure 1B reveals a clear peak in the δ/theta-range at P40 which is not present in other groups.

We compare the data set corresponding to spontaneous activity and optogenetically-driven activity and did not detect a shift in the frequency of slow (i.e. δ/theta) oscillations. The misleading impression of differences between the two conditions in this frequency range might results from the fact that Figure 1B and Figure 3B show individual traces and power spectra or its modulation index. Only Figure 1C,D and Figure 3C,D summarize the full set of recordings. We changed Figure 1C and Figure 3C (z-scaled results per age) to highlight the increase in high-frequency activity, which is the topic of the present study.

We agree with the reviewer that the spontaneous high frequency activity appears to be more broadband than the optogenetically-driven one. This might result from the fact that activation of 25-30% layer 2/3 pyramidal neurons with non-specific ramp stimulations is most likely one but not the unique mechanisms of γ generation in the developing mPFC. However, the similar dynamics supports the suggestion that optogenetically-driven activity reflects at least certain aspects of spontaneous activity. We added a discussion of the relationship between spontaneous and optogenetically-driven fast activity to the manuscript (Discussion section).

As suggested, we tested for significant correlations of peak frequency and amplitude for baseline and ramp periods at the level of individual recordings. Both, peak frequency and amplitude between baseline and ramp stimulation are significantly correlated. We added the results to the Results section and addressed the issue in the Discussion section. To disentangle the changes in peak frequency vs. power increase, we identified the best set of predictors of the two parameters using stepwise linear regression. We found that, while peak frequency correlated with fast-spiking properties and single unit rhythmicity, peak amplitude was best predicted by optogenetically-driven firing rate changes, indicating an important effect of stimulation efficacy on peak amplitude. We included the new results of analysis in the text (Discussion section).

ii) Reviewers commented that there is very little low frequency power for the P5-10 group. Reviewers would like to see what happens if the authors normalize the power differently (using a standard division by the total power), and if there would not be more γ power at high frequencies?Related to this, reviewers commented that the spontaneous firing rates seem to be very low for the P5-10 group (see Figure 4) and were concerns that comparing absolute power across animals amounts to comparing mean firing rates.

Cortical LFP power is generally low in young mice, due to low and discontinuous activity. In this manuscript we focus on the development of γ activity and therefore chose a scaling that enables optimal visualization of γ band activity. The development of theta activity in the mPFC has been previously addressed in several studies of our lab (Brockman et al., 2011; Bitzenhofer et al., 2015; Hartung et al., 2016). Figure 4 c-e shows firing rate changes, not absolute firing rates. Low firing rates (together with low rhythmicity and network coupling) in neonatal cortex most likely contribute to low LFP power at P5-P10. Further, we see a similar increase in γ power and frequency induced with ramp light stimulations, where we compare power change (modulation index) instead of absolute power (Figure 3).

iii) The lower peak seems to shift to a lower frequency with age, rather than to a higher frequency. The authors should discuss this.

The power spectra shown in Figure 1B are examples of power spectra of 4 individual recordings. We clarified this issue in the figure legend and changed the plot summarizing all data in Figure 1C for better illustration. Instead of the average power spectra, we show z-scored average power spectra for each age to increase the visibility of low power levels at young age.

iv) There appear to be artifacts in the spectrum shown in Figure 1, column 3, second row P12 – what are these artifacts? Artifacts are peaks separated by about 10Hz, meaning they are 0.1s artifacts, this could be due to anesthesia apparatus? Could this have influenced the quantification of γ?

The artifacts are not due to anesthesia apparatus, since mice have been anesthetized through intraperitoneal injections of urethane. In line with previous results, the artifacts might mirror the breathing of pups. Due to the thin and instable skull bone these artifacts are difficult to avoid, especially in non-anesthetized animals. They appear rather large in the spectra because the LFP power level is low. We analyzed the data and confirmed that in all but one pup, which we excluded from the analysis, the artifacts did not preclude reliable peak detection.

v) Figure 3C: From this figure, the developmental change in spectral power is not clearly visible. The authors may want to use a different way of highlighting changes in spectral power across development.

In line with the reviewer’ suggestion, we changed the plots in Figure 3C, Figure 3—figure supplement 1B, and Figure 1C by z-scoring the data for each age to increase the visibility of lower power levels/changes at young age.

4) Effect of behavioural state. The authors should address the issue of behavioural state.In general comparisons are made between animals where it is not clear that the behavioural state is the same. For example, chronic recordings were done only in P25 and P40 groups, but the behavioural state of the younger (and the conditions of recordings) during wakefulness are completely unclear. It is not clear whether changes in γ observed might be due to behavioural state (see also Hoy and Niell, 2015). Was the behavioral state the same between mice? Did mice at different ages run more than at other ages? How did running impact FS or RS firing?

We agree with the reviewer on the high relevance of behavioral state. For this, we added the Supplementary file 1 and rephrased the text for clarification of recording conditions. Out of 115 recordings considered for the present study, 80 recordings were performed in P5-40 mice under anesthesia with acute head fixation (1 recording per mouse), 10 recordings were performed in P7-12 mice without anesthesia with acute head fixation (1 recording per mouse), and 25 recordings were performed without anesthesia with chronically implanted head fixation (6 mice at P24-26 and 5 mice at P37-40). All recordings were performed from head-fixed mice using acutely inserted electrodes.

We previously investigated in detail the patterns of oscillatory activity in the presence of anesthetics and their relationship to sleep (Chini et al., 2019). Mice at young age spent a large portion of time sleeping and no significant differences in the activity patterns recorded up to 1 hour were detected. Their poor motor abilities at this age made an investigation on a spinning disk superfluous. Only non-anesthetized mice at older age (P24-26 and P37-40) with chronic head fixation implants were free to sit or run on a spinning disc. Running increased γ power in these mice. To reduce the variability due to different behavioral states, recordings from non-anesthetized mice with chronic head fixation implants were excluded from single unit analysis.

The reviewers further wondered how the time in the session impacted likelihood of seeing different rhythms (did lower frequencies predominate later in the session, as many prior studies would predict)?

Recordings followed a specific protocol, such that the time in the session does not affect the results. As exemplified in Figure 4A,B and Figure 4—figure supplement 1 the spontaneous activity and optogenetic activation of individual units across trials was rather consistent. An in-depth investigation of modulatory factors of γ activity, such as anesthesia depth, tiredness, stress exceeds the aims of the current manuscript and will be the topic of future studies.

Overall, the authors feel that the data in this sense is underused and that there may be a lot of useful developmental richness that would make the paper a more important contribution.

We agree with the reviewer that the data contain further interesting information about the development of prefrontal activity, that may be addressed in future studies. We decided to focus, in the present manuscript solely on the development of fast oscillatory rhythms in the mPFC. To backup this goal, we exploited the richness of data to perform additional analyses addressing rhythmicity of single units and locking of single units to LFP. The new results have been added to the manuscript (subsection “Fast spiking interneuron maturation resembles the time course of γ development”). Of note, the development of slower oscillations in the mPFC has been addressed in previous studies (see point 3 (ii)).

The methods say some wake animals were recorded from a stereotaxis apparatus. What does this apparatus imply for behavioral state?

We developed a novel recording paradigm aiming to monitor the cortical activity of young mice. Due to the size of the animals as well as fragility and growth-induced changes of the skull, the “classical” tetrodes or telemetric recordings cannot be used for this purpose. Therefore, we built a moving disk on which non-anesthetized head-fixed mice are free to move. To improve the clarity of Materials and methods section, we rephrased the text and added a Supplementary file 1 summarizing the experimental conditions. We added to exemplary images to illustrate the two recording condition below:

The conditions of the recordings need to be clearly described.

As requested, we rephrased the text (Materials and methods section, Results section, figure legends) and added Supplementary file 1 summarizing experimental conditions.

5) Richer analyses of the adaptation functions is highly warranted, given interesting work (e.g., by Llampl and many others) in the adaptation of inhibitory versus excitatory currents.Reviewers further wondered if the difference in apparent adaptation cannot be simply explained by the overall difference in activation to light trains?

We performed multifactorial ANOVA with the main factors RS/FS, age group, and stimulation frequency to compare adaptation functions and added the results to the figure legend and Supplementary file 2. RS and FS units were not significantly different, but adaptation functions changed significantly with age and stimulation frequency. Stronger adaptation at young age and high frequencies likely reflects an inability of neurons in the mPFC of neonatal mice to spike at high frequencies (see also Bitzenhofer et al., 2017). It is not clear to us how lower activation by light trains would explain a change in the ratio from early to late activation. We suggest that the lower activation in response to pulsed light for young age groups mainly reflects reduced connectivity and thereby reduced activation of not-opsin-expressing neurons.

6) While FS units are generally assumed to correspond to PV interneurons, pyramidal neurons (Onorato et al., 2019) and Somatostatin interneurons (Ma et al., 2006) can also display these characteristics. Thus, the FS units identified in this study might not only include PV interneurons. The authors should discuss this.

We added a brief discussion of the topic suggested by the reviewer to the Discussion section.

The authors wondered if the authors can give a strong argument or piece data that the "fast-spiking firing properties" are also altered, in the sense of the neurons' input-output functions (how they respond to current or synaptic input). That is fast spiking needs to be more clearly defined and interpreted.

In line with the suggestion, we refer to previously published data that characterized the development of PV interneurons in the mPFC in vitro (Miyamae et al., 2017).

7) In general reviewers agreed that the dataset (nature of the statistical units, mice, recording and state – awake, anesthetized, sex) is not described clearly enough. These points would greatly improve the paper.i) Reviewers noted it is generally difficult to figure out how the authors exactly analyzed their data, what data points went in the figures, etc. Tables summarizing the dataset would be very useful.

We added the requested information to the text (Materials and methods section, Results section, Figure legends) and summarized the experimental conditions in a new table (Supplementary file 1).

ii) Chronic recordings were done only in P23-25 and P38-40 – are these difference mice?Figure 1C – are these different mice? If so, can we compare them to each other? What data goes exactly in this figure in terms of acute recordings and chronic recordings for Figure 1A-C. It seems for the wake group the authors compare chronic to non-chronic conditions. That comparison seems problematic given the effect on behavioural state.

As mentioned above, we rephrased the text to improve the clarity of our statements and added the Supplementary file 1 summarizing experimental conditions. Especially, we aim to specify that the “chronic recordings” in young mice are not similar to those performed in adults but correspond to acute recordings in mice with chronically implanted head fixation adapters. Each of these mice was recorded 1-3 times either between P24-26 or P37-40. Thus, all recordings were acute head-fixed recordings with electrodes inserted prior to the recording session, but 11 mice at P24-26 and P37-40 had chronically implanted head fixation adapters. These 11 mice were added to show that the developmental increase in γ frequency does not depend on anesthesia. They were excluded from the analysis of single units to avoid the comparison of different behavioral states.

iii) N = 114 recordings, but n=80 and n=20,35 in the figure legends what does it refer to? The n's used for the analysis and the number of mice should be clear everywhere in the paper. Is every mouse contributing a separate data point for the statistics, or is every channel, or every recording a data point? If so, isn't the analysis problematic because you are inflating your statistical degrees of freedom massively. For example, p=2.73*10-8 seems to be a p-value that is suffering from inflation due to dependent measures. Similar comments apply to the rest of the paper.

We thank the reviewer for highlighting the error in figure legend of Figure 1. For Figure 1, the total was n=114 recordings, 80 recordings from anesthetized mice (one per mouse) and 34 recordings from 20 non-anesthetized mice that underwent multiple recordings (1 non-anesthetized recording was removed from baseline LFP analysis because recording artifacts interfered with γ peak detection). For Figure 3, the total was n=115 recordings, 80 recordings from anesthetized mice (one per mouse) and 35 recordings from 21 non-anesthetized mice. We corrected the mistake. Throughout the manuscript, we aimed to clarify the sample size and corresponding statistics by rephrasing the text and adding the Supplementary file 1 summarizing the experimental conditions.

It is unclear how sample size impact the inferences we should take from different plots and analyses--many more cells are sampled at certain days. For example, in Figure 6B, there seem to be differences in the adaptation functions, but it's not clear whether this is the case.

Sample sizes are low for FS at young age groups because of the late maturation of FS interneurons. We added statistical comparisons throughout the manuscript to identify statistically significant differences. Low numbers of FS at young age groups prevent strong conclusions.

iv) Figure 1B panel right -there are no error bars on this figure, how reliable are these differences across mice? Figure 1 Heatmap – it seems that there is a lot variability, perhaps across different mice? The authors should show how consistent the effects are across mice that are recorded under similar conditions.

The lack of error bars is due to the fact that Figure 1B panel right shows example power spectra of 4 individual recordings. The full dataset is displayed in Figure 1C and D. We modified the figure legend for better clarity. Moreover, we replaced the plot summarizing all data in Figure 1C as average power spectra by z-scored average power spectra for each age. Each data point in Figure 1D represents an individual recording, showing their variability, but a consistent increase of γ frequency across age.

v) The authors should describe the intensity of the laser, and say how the intensity was calibrated and determined. The reviewers wondered whether the laser is actually driving activity in the pups, because this is not clearly visible from the plots, perhaps due to extremely low spontaneous firing rates. It appears that high frequency activity (which is dominated by spiking) does not seem to be affected by the laser. Single examples of neurons with raster plots would be very helpful to understand this. In general, providing raster plots with single spikes would be extremely helpful for the reader to assess the reliability of optogenetics simulation and the meaning of various statistics computed. The authors should ideally show an example of a single neuron that is positively modulated by light stimulation for the young pups to understand what is going on.The text should clearly distinguish between absolute rate changes and modulation. If spontaneous firing rates are very low, then neurons might show a large relative change, but they could still be largely unresponsive to the light.

In line with the reviewer’s suggestions, we added a description of laser intensity calibration to the Materials and methods section. We replaced the plot in Figure 3C showing the modulation index of induced power by the z-scored modulation index for each age to enable a better visualization of effects at young age. Induced activity is lower in young mice, presumably due to lower connectivity and weaker synchronization of non-opsin-expressing neurons.

We added raster plots of example RS and FS units at different ages positively modulated by optogenetic stimulation to Figure 4 as well as example units that were inactivated or not affected by the stimulation from the same recordings to Figure 4—figure supplement 1.

We rephrased the text to avoid confusions about absolute rate changes and modulation.

vi) The following statement is not clear. The first p-value doesn't seem to be significant."While ramp-induced firing rate changes of RS units (Mann207 Kendall trend test, p=0.07, n=7 age groups, tau-b 0.619) became more prominent at older 208 age, the firing rate changes were stable for FS units (Mann-Kendall trend test, p=0.88, 209 n=7 age groups, tau-b 0.047) (Figure 4B)"

We rephrased the sentence.

vii) "whereas at older age the number 211 of activated and inactivated RS units got more balanced (Mann-Kendall trend test, p=1.52*10-14 212 , n=1821 units, tau-b -0.123)"Please explain what "more balanced" means and how balanced is quantified.

We rephrased the sentence.

viii) Reviewers commented that in Figure 5, it is unclear what the quantification or statistical test is. It looks like only the P36-40 is qualitatively different, and the P5-10. But the intermediate ages do not seem to show a consistent pattern.

To address the query, we performed multifactorial ANOVA with the main factors RS/FS, age group, and frequency to compare the autocorrelation power. We added the results to the text (Discussion section) and Supplementary file 2.

viiii) A variety of methods are available for hierarchical clustering. However, a reviewer did not understand which of these the authors use for the clustering of single units. Please make sure it is well detailed in the Materials and methods sections.

We specified that Pairwise Euclidean distance was used for hierarchical clustering (subsection “Fast spiking interneuron maturation resembles the time course of γ development”).

Comments on literature:i) The authors highlight in this paper the importance of early development for the maturation of neural oscillations. It would be useful to expand the literature review to indicate that development of rhythmic activity as well as the underlying mechanism extend beyond P40.

According to our data, peak frequency and amplitude of γ activity stabilize around P25 and no major changes occur afterwards. We added a brief discussion of the γ activity and underlying mechanisms in mice beyond P40 (typically considered as young adult) (Discussion section).

ii) In the Introduction, please specify that the results obtained by Chen et al., 2017; Veit et al., 2017 were acquired in the visual cortex as the type of β activity observed in these studies has not yet been observed in other regions.

We rephrased accordingly.

iii) Discussion and comparison with existing literature (Hoy and Niell, 2015 – which was not cited) is necessary. That study contains several findings reported here as well.

We included the mentioned paper in the Discussion section.